# Epstein-Barr Virus Encephalitis: A Review of Case Reports from the Last 25 Years

**DOI:** 10.3390/microorganisms11122825

**Published:** 2023-11-21

**Authors:** Marine Peuchmaur, Joris Voisin, Mathieu Vaillant, Aurélie Truffot, Julien Lupo, Patrice Morand, Marion Le Maréchal, Raphaele Germi

**Affiliations:** 1University Grenoble Alpes, CNRS, DPM, 38000 Grenoble, France; marine.peuchmaur@univ-grenoble-alpes.fr; 2University Grenoble Alpes, CNRS, CEA, IRIG IBS, 38000 Grenoble, France; joris.voisin@gmail.com (J.V.); atruffot@chu-grenoble.fr (A.T.); jlupo@chu-grenoble.fr (J.L.); pmorand@chu-grenoble.fr (P.M.); 3University Grenoble Alpes, Service de Neurologie, CHU Grenoble Alpes, 38000 Grenoble, France; mvaillant@chu-grenoble.fr; 4University Grenoble Alpes, Inserm, CHU Grenoble Alpes, GIN, 38000 Grenoble, France; mlemarechal@chu-grenoble.fr

**Keywords:** Epstein-Barr virus, encephalitis, biology, imaging, treatment

## Abstract

Although uncommon, Epstein-Barr virus-related neurological disorders represent the seventh most frequent cause of infectious encephalitis in adults. The limited number of publications on EBV encephalitis mainly document isolated clinical cases. This study aimed to summarize published data on EBV encephalitis. A systematic literature search identified 97 EBV encephalitis cases. In the selected cases, EBV-related neurological disorders manifested as lymphocytic pleocytosis in the cerebrospinal fluid (CSF) with moderate hyperproteinorachia. The EBV PCR test was positive in 87% of the CSF samples, with wide-ranging viral loads. When encephalitis occurred in the context of past EBV infections, all of the EBV PCR tests on CSF samples were positive. On the contrary, negative EBV PCR tests on CSF samples occurred only in the context of primary infections. EBV PCR was rarely carried out on blood samples, contributing minimally to the diagnosis. For the treatment of EBV encephalitis, Aciclovir was used alone in 29% of cases, and in association with other drugs in 40% of cases. Ganciclovir (30%), corticoids (52%), and immunoglobulins (15%) were mainly used in association with other drugs. Cerebral imaging was abnormal in 69% of cases, mostly in the cerebellum and basal ganglia. This work highlights that the EBV PCR test on CSF samples is currently the main laboratory diagnostic test to diagnose EBV encephalitis. This diagnostic test is useful; however, it is imperfect. New complementary diagnostic tools, approved treatments, and standardized practices could improve patient management.

## 1. Introduction

The Epstein-Barr virus (EBV) is a highly prevalent virus, with more than 95% of the population infected worldwide. Identified in 1964 in tumor cells, the EBV is the first infectious agent linked to cancer. The EBV belongs to the gammaherpesvirus subgroup of the *Herpesviridae* family. Primary EBV infection is usually asymptomatic when occurring in childhood, but it induces infectious mononucleosis in adolescence or adulthood. The EBV is usually transmitted by contact with oral secretions. It can also be transmitted by organ or hematopoietic stem-cell transplantation. The incubation period of infectious mononucleosis is approximately 6 weeks. Its symptoms are typically self-limited and resolve within several weeks, varying with fever, fatigue, pharyngitis, hepatosplenomegaly, and cervical lymphadenopathy [1]. After recovery from an acute infection, the virus establishes a long-term latent infection in B-cell lymphocytes, with possible recurrent reactivations and saliva excretion that are normally limited in immunocompetent hosts, due to their efficient cytotoxic cellular immunity. The key to the establishment of latency in B cells is the virus’ ability to induce continuous growth and cell transformation. This property also implies that this pathogen is a potent oncogenic virus. Given its B-cell tropism, EBV infection can be associated with B-cell lymphomas such as Hodgkin’s and Burkitt’s lymphoma, as well as with lymphoproliferative disorders observed in the context of immunodeficiency, and particularly in the context of transplantation [2]. The EBV is also associated with primary central nervous system (CNS) lymphoma, a highly aggressive non-Hodgkin’s lymphoma that is limited to the CNS and that mainly occurs in immunocompromised hosts (HIV/AIDS, transplant), but also in immunocompetent patients [3,4]. The EBV also presents epithelial cell tropism and is associated with epithelial cancers such as nasopharyngeal carcinoma.

The *Herpesviridae* family is well known for its CNS pathogenicity, mainly represented by herpes simplex encephalitis type 1, followed by varicella zoster virus encephalitis. Despite its scarcity, EBV encephalitis was described as the third etiology of *Herpesviridae* encephalitis and the seventh etiology of infectious encephalitis [5]. Indeed, the EBV has long been known to cause neurological damage, as the first case described following infectious mononucleosis dates back to 1931 [6,7]. Since then, EBV infections have been found to lead to various neurological manifestations of varying severity, such as encephalitis, meningitis, myelitis, cerebellitis, polyradiculitis, Guillain–Barré syndrome, Alice in Wonderland syndrome, and more recently, multiple sclerosis. However, these disorders are particularly rare, and only a few isolated clinical cases have been published. Moreover, the molecular mechanisms involved in CNS EBV infection have not yet been elucidated, and contrary to other herpes viruses, such as herpes simplex and varicella zoster, the EBV has not been shown to achieve latency in neurons [3,4].

Due to the infrequency and diversity of the neurological complications related to EBV infections, there is currently no comprehensive and detailed description of the EBV. Only a few retrospective studies, generally based on single-center trials or targeted populations such as pediatric patients, are available [8,9]. This review aimed to gather data from individual reported cases of EBV encephalitis based on epidemiological, clinical, biological, or therapeutic factors.

## 2. Materials and Methods

A systematic literature search for clinical cases reporting EBV infections of the CNS was performed on the PubMed database using different combinations of the following keywords: (“Epstein-Barr Virus” or “EBV” or “Herpesvirus 4, Human”) and (“Encephalitis” or “Meningitis” or “Meningoencephalitis” or “Myelitis”) and “case report”. The reviewed articles were published in English, French, German, Spanish, Portuguese, and Norwegian over the last 25 years. Publications without biological data were excluded. Following this methodology, 104 clinical cases from 98 articles were selected [10,11,12,13,14,15,16,17,18,19,20,21,22,23,24,25,26,27,28,29,30,31,32,33,34,35,36,37,38,39,40,41,42,43,44,45,46,47,48,49,50,51,52,53,54,55,56,57,58,59,60,61,62,63,64,65,66,67,68,69,70,71,72,73,74,75,76,77,78,79,80,81,82,83,84,85,86,87,88,89,90,91,92,93,94,95,96,97,98,99,100,101,102,103,104,105,106,107].

## 3. Results

### 3.1. Epidemiological Analysis

The 104 cases were reported from all over the world but mainly from Europe and Asia, with 43 and 36 cases, respectively. Table 1 provides an exhaustive list of the countries of origin of each case with the associated article references.

Among the 104 patients with EBV-related neurological complications, 58% were male. The median age of patients was 22 years of age, with the youngest patient being 10 months of age and the oldest 77 years of age (Figure 1). Overall, 46% of subjects were under 20 years of age, 45% between 20 and 60 years of age, and only 9% over 60 years of age. Among those under 20 years of age, 26% were children (0–10 years of age), and 20% were adolescents or young adults (10–20 years of age).

The vast majority of patients (83, 80%) were immunocompetent. Of the twenty percent of immunocompromised cases, six involved HIV-positive patients [11,16,38,44,45,100], including three cases with a CD4 count below 250/mm^3^. Nine cases were related to kidney transplant recipients [12,15,27,33,42,61,68,72,103], who were mainly treated with tacrolimus and mycophenolate mofetil (eight patients) with or without corticoids. The time until the appearance of neurological symptoms varied from several months to more than 10 years after transplantation. In four transplant cases, the EBV serological statuses of the donors and recipients were specified: two involved EBV-seropositive donors and EBV-negative recipients, and two concerned EBV-seropositive recipients. Among the remaining immunocompromised patients, two underwent hematopoietic stem cell transplantation: one with an EBV-seropositive donor and recipient and one was not specified [46,77]. One immunocompromised case presented with inflammatory bowel disease, which was treated with methotrexate and prednisolone [101]. Another suffered from Vaquez disease, which was treated with hydroxycarbamide [30]. Finally, two immunocompromised patients were undergoing chemotherapy for breast cancer and radiochemotherapy for lung adenocarcinoma [81,101].

### 3.2. Clinical Observations

The majority of cases with EBV-related neurological disorders involved encephalitis (sixty-seven), sometimes associated with meningitis (eight), myelitis (three), or radiculitis (one). Cerebellitis affected seventeen percent of patients (eighteen, including one case with associated meningoencephalitis classified as such), while isolated meningitis impacted only six percent of cases (Figure 1). The remainder of this study will focus on the 97 cases of encephalitis or cerebellitis.

At the time of diagnosis, clinical data were classified into two subcategories: general symptoms and neurological symptoms. General symptoms were quite similar and unspecific. Patients complained most often about headache and/or fever (83%) or digestive disorders such as nausea, vomiting, and diarrhea (40%). Less frequent symptoms included asthenia (14%), pharyngitis or tonsillitis (6%), urinary disorders (6%), myalgia (6%), and lower back pain (2%).

The most frequently observed neurological disorder was confusion (35%), including impaired consciousness, disorientation in time, place, or person, delirium, and visual hallucinations. Cerebellar syndrome was the second most common neurological symptom (30%), which manifested as gait disorders or unsteadiness, followed by generalized tonic-clonic seizures (26%) and meningeal signs (16%) such as neck stiffness. Other symptoms were less frequent: extrapyramidal syndromes (14%), including abnormal movements or parkinsonian syndromes, and psychiatric conditions (12%), including anxiety, anhedonia, personality disorders, irritability, and inappropriate behavior. Neuromuscular disorders (11%), such as hypotonia, myalgia, and spasticity, were also described, as were visual sensory impairments (10%). Cognitive disorders (7%), like short-term memory impairment and vertigo (6%), were less frequent. Lastly, sensory disorders, such as neuropathic pains, paraesthesia, and hypoesthesia, were cited in a few cases.

Mortality was observed in fifteen patients (15.5% of all cases) including four immunocompromised patients (four of fifteen patients = 26%) within 5 days to 4 months of diagnosis. The causes of death were generally linked to complications occurring in the context of coma, tetraplegia, multisystem organ failure, edema, and brain hemorrhage.

Excluding the 15 patients who died and 12 patients for whom no follow-up data were available, 20 out of 70 patients (29%) who benefited from medium- or long-term follow-up experienced different types of disabilities. Among those presenting with sequelae, their type and intensity varied, involving cognitive, sensory, muscle, balance, and language disorders. For instance, some patients retained memory impairment, hearing loss, residual diplopia, or psychomotor retardation. It is noteworthy that follow-up was not standardized in the publications.

### 3.3. Biological Data

#### 3.3.1. Cerebrospinal Fluid and Brain Analysis

Among the 97 selected cases, polymerase chain reaction (PCR) tests for EBV DNA detection in the cerebrospinal fluid (CSF) were performed in 70 patients (Appendix A). For the 27 patients who did not benefited from EBV PCR tests on their CSF samples, the diagnosis of EBV encephalitis was based on blood virological testing, such as EBV serology or EBV PCR tests. Among the 70 patients for whom CSF samples were analyzed with EBV PCR tests, sixty-one were positive (87%) and nine were negative (13%). Most EBV PCR tests on CSF samples were qualitative (43/70, 61%), whereas only 27 PCR tests (38%) were quantitative (qPCR). The viral loads ranged from 68 to 81,200 copies/mL, with a median value of 2400 copies/mL (IQR 14,173 copies/mL). For the nine patients with a negative EBV PCR test from a CSF sample, the diagnosis of EBV encephalitis was based on one of the following tests: (i) the presence of an anti-EBV IgM test highlighting primary EBV infection (six patients); (ii) a positive EBV qPCR test from a brain biopsy sample 98 days after stem-cell transplantation (one patient); (iii) a positive EBV PCR test on a sample from a second lumbar puncture procedure, performed after 12 or 16 days (two patients).

Indeed, for 19 patients, a second CSF sample was collected 5 to 90 days after the first CSF sample (median 20 days). In seven cases (37%), the second CSF sample did not contain EBV DNA, but in 12 cases (63%), the EBV viral load was positive. As described above, in two of these cases, the first CSF sample was negative for the EBV. For the remaining ten cases, two had a lower CSF viral load than the viral load that was collected at diagnosis, five had a similar viral load, and three had a higher viral load. Four out of the nineteen patients died (21%): two had a negative viral load and two had a similar viral load for both samples.

Brain biopsies were reported for ten patients, including three post-mortem biopsies [27,30,37,38,43,44,54,56,77,103]. In five cases, the analysis aimed to exclude cerebral lymphoma. The EBV was detected in all brain tissue samples: five cases with PCR tests, three cases with immunohistochemical tests, and two cases by positive in situ hybridization.

EBV serology tests on CSF samples were performed in only seven cases and poorly documented. The EBV serology test on CSF sample was negative in one patient with a primary EBV infection, and positive in six cases: one with a primary EBV infection and a positive EBV PCR test on a CSF sample, three with a past EBV infection for which the diagnosis was confirmed by the presence of anti-VCA IgG in the CSF, and two with unavailable EBV serology tests on serum samples.

White blood cell (WBC) counts in the CSF were determined for 79 out of 97 cases (81%) (Figure 2 and Table 2). Among them, pleocytosis was observed in 66 cases (83%). The median WBC count was around 47/mm^3^ (IQR 97/mm^3^), but 60/mm^3^ when only including abnormal WBC counts > 5/mm^3^. The highest WBC count was 797/mm^3^, but most cases presented mild pleocytosis (30% less than 20/mm^3^). For 46 patients, the cell population was specified: 87% of lymphocytic pleocytosis and 13% of neutrophilic pleocytosis.

Proteinorachia was measured in 86 out of 97 cases (82%). Considering a physiological value of less than 0.4 g/L, 60% of patients presented an elevated concentration of protein in the CSF. In most of the 63 cases with quantitative data, a slight to moderate increase was recorded, with more than 57% with proteinorachia less than 1 g/L (Figure 2). The median proteinorachia was 0.84 g/L (IQR 1.05 g/L) (and 1.05 g/L if only hyperproteinorachia is included), and the highest value reached was 4.1 g/L.

When concentrations of protein in the CSF and cell content analyses were both documented (77 cases), 46 cases (60%) had both hyperproteinorachia and pleocytosis, whereas eight cases (10%) presented neither. Among these eight cases, three were labelled as primary EBV infection, one was labelled as recurrent/chronic EBV infection, EBV PCR tests on CSF samples were positive in two cases, EBV serology test on CSF sample was positive in one case, and finally, one was poorly documented. 

#### 3.3.2. Blood Analysis

EBV serology tests were available in 75 out of 97 cases (77%). Primary infection was characterized for 36/75 patients (48%) due to the presence of anti-VCA IgM +/− anti-VCA IgG, or due to negative serology tests with positive EBV PCR tests on CSF or blood samples. Past EBV infection, determined by the detection of anti-VCA and/or anti-EBNA IgG without anti-VCA IgM, was confirmed in 21 cases (28%). For the 18 remaining cases (24%), the incomplete data prevented us from drawing conclusions.

Among the 36 cases of primary infection, EBV PCR tests on CSF samples were performed for 28 patients, and were positive for 21 patients (75%). Among the seven cases (25%) with negative EBV PCR tests on CSF samples, one patient underwent a second lumbar puncture procedure after 16 days, which was positive. No other data were provided for the remaining patients. When encephalitis occurred in the context of a past EBV infection, all of the EBV PCR tests on CSF samples were positive (Table 3).

EBV PCR tests on blood samples were only described for 23 cases (24%), 16 of which were positive. This analysis was mainly performed for primary infection cases (11/23, 48%), with nine being positive (81%). EBV PCR tests were performed in only six of the identified past EBV infections, four of which were negative (66%). Among them, three had a positive EBV load in the CSF.

Of the 16 positive EBV PCR tests on blood samples, nine were qPCR tests, with the viral load ranging from 78 to 560,000 copies/mL (some viral load results were not given in copies/mL and were thus not regarded as quantitative), with a median value of 2270 copies/mL (Figure 2).

Cell blood counts were available for 54 out of 97 patients (56%), and leukocytosis was observed for 20 patients (37%). In most cases, leukocytosis was mild, with a maximum value of 25 G/L (median 6.5, IQR 7.3 G/L). The cell population was described for only nine patients: six presented with neutrophilic polynucleosis, while three had lymphocytosis. 

C-reactive protein (CRP) levels were measured for 26 out of 97 encephalitis cases (27%) and were mostly normal with a median value of 1 mg/L (IQR 24 mg/L). Indeed, 16 patients (61%) had a CRP value less than 5 mg/L. For the 10 patients with elevated CRP levels, the median value was 92 mg/L, and the highest value was 429 mg/L.

### 3.4. Imaging

Electroencephalograms (EEGs) and medical imaging techniques such as magnetic resonance imaging (MRI) scans, computerized tomography (CT) scans, and even positron emission tomography (PET) scans are commonly used for the diagnosis, prognosis, and monitoring of encephalitis. In the 97 cases, 88 (91%) mentioned MRI scans. EEGs were recorded in 40 cases (41%). EEG findings were normal for seven patients, with the most frequent abnormality being diffuse slowing without discharges (19/40).

When MRI scans were performed, encephalitis-related abnormalities were found in 69% of patients (61/88). In 41% of cases (25/61), the abnormalities affected multiple anatomical regions. Even though none were specific to EBV encephalitis, some regions were mentioned more frequently, such as the cerebellum, basal ganglia, frontal lobe, and cerebellum (Table 4).

Among the 27 patients who underwent both a brain CT scan and MRI scan, 12 (44%) had consistent reports for both types of imaging, whereas 14 (52%) had an MRI scan with encephalitis-related abnormalities and a normal CT scan (Table 4).

### 3.5. Treatments

For 18 of the 97 cases, treatment was not reported (either no treatment or no data).

Regarding the 79 remaining patients, acyclovir was prescribed in 55 (70%) cases. In most cases, aciclovir was used as empirical treatment for presumed herpes simplex meningoencephalitis. Acyclovir was mostly maintained following the diagnosis of EBV encephalitis, even though medical societies do not recommend its use for this indication. In 11 cases, acyclovir was stopped: four patients aged 2, 6, 10, and 24 years evolved positively; acyclovir was replaced by (val)ganciclovir in four patients, but two died and two recovered with neurological sequelae; and in three cases, acyclovir was stopped but quickly reintroduced due to worsening, with one resulting in death.

Most of the 55 patients receiving acyclovir presented a positive evolution, although 10 died. Among the 23 patients treated with acyclovir alone, only one died. The mortality rate of patients treated with combined therapy was higher (9/34, 26%). The drug most often combined with acyclovir was corticoids: 19 patients, including three deaths. 

The second most common antiviral therapy was (val)ganciclovir with 24 cases (30%). Ganciclovir was administered alone in six cases. In five cases, it was combined with corticoids, with two of these patients dying. In one case, ganciclovir was stopped after 14 days and switched to foscarnet, which was in turn replaced by brivudine after 7 days due to the increasing CSF EBV load. Overall, seven patients (29%) treated with ganciclovir died, all of whom received a combination of drugs.

Corticosteroids were given to 42 patients (53%), nine of whom received them for monotherapy. Overall, nine of the 42 patients died (21%), one of whom received only corticosteroids.

Antiviral drugs were frequently combined with corticoids and/or intravenous immunoglobulin (IVIg) (Table 5). Irrespective of the combination therapy, deaths were observed, except for the simultaneous use of ganciclovir, IVIg, and corticoids (and the combination of IVIg and corticoids, which was only administered to one patient).

In some rare cases, other drugs or combinations were used. For instance, rituximab was used in combination for six patients, two of whom died. Two patients treated with acyclovir, ganciclovir, and foscarnet both died. Plasmapheresis was used for three patients with a successful outcome.

For patients who underwent a kidney transplant, the immunosuppressive therapy was decreased with no graft rejection or death, except for one patient.

## 4. Discussion

EBV encephalitis is one of the third to fifth most frequent causes of viral encephalitis, depending on the study. However, our literature search identified 97 clinical cases describing EBV-related encephalitis. This work aimed to study and summarize these rarely published cases.

The published cases of EBV encephalitis come from all over the world, with a majority in Asia and Europe. It is not surprising that less developed countries in Africa and South America publish less on this subject, although the limited number of publications from North America might suggest a lower incidence of EBV encephalitis in this region of the world.

The patients were essentially immunocompetent and young subjects (in agreement with the number of cases of primary EBV infection) which is comparable to all-cause encephalitis cohorts such as the cohort of Granerod et al. [108] in England involving 203 patients. In all these patients, the clinical manifestations associated with EBV encephalitis were acute in onset and not very specific when compared with other studies of encephalitis cohorts [5,108,109].

Imaging was also not very specific and revealed a heterogeneous topography of brain damage, with a tendency to affect the cerebellum, basal ganglia, and frontal lobe more frequently. MRI scans were abnormal in 69% of cases, and in more than half of cases, there were discrepancies between the CT scans and MRI scans, with an abnormal MRI scan but normal CT scan. This discrepancy highlights the necessity of performing a brain MRI scan for patients presenting with EBV encephalitis. No localizations were specific to EBV encephalitis.

Biological data were similar to those described in viral encephalitis of other etiologies: lymphocytic pleocytosis (87%), accompanied by moderate hyperproteinorachy Ref. [5]. In rare cases, pleocytosis may be absent or have a majority of polynuclear cells. Blood biology characteristics such as blood counts and CRP levels were difficult to interpret in our study. Often, these characteristics were not mentioned in publications, and when they were included, they were normal in more than 50% of cases and thus contributed very little.

EBV PCR tests on CSF samples were the most frequent tests performed after WBC tests and tests for protein concentration in the CSF, and were positive in the vast majority of cases (87%).

Thus, EBV PCR tests on CSF samples appear to be crucial for the etiological diagnosis of EBV encephalitis. EBV PCR tests on brain biopsy samples are also relevant, since all 10 biopsies were positive for the EBV. However, the small number of cases does not allow us to draw definite conclusions about the diagnostic performance of this analysis, while the invasive and risky nature of the brain biopsy means that it cannot be considered as the primary diagnostic test. It is noteworthy that 30% of the biopsies were sampled for suspected cerebral lymphoma.

The viral loads measured in the CSF did not allow us to determine a threshold for the association with the disease. Indeed, viral loads showed a wide variation, and low viral loads (below 1000 copies/mL) were retained for diagnosis [31,34,41,43]. Many EBV PCR tests were qualitative, and when they were quantitative, the methods were not standardized or specified, thus making them difficult to compare.

To complete the discussion on EBV PCR tests on CSF samples, our virological practice shows that the presence of the EBV in the CSF is not necessarily exclusive to EBV encephalitis, and that a positive intrathecal EBV load can be observed in many cases with another etiology. Positive EBV PCR tests on CSF samples are therefore not specific to EBV encephalitis, especially in cases of low viral load. This point may require further study.

EBV PCR tests on blood samples were rarely included in the published cases, probably because presence of the virus in the blood is not specific to the EBV encephalitis. Indeed, people with a high viral load in the blood do not necessarily have EBV encephalitis, while those with EBV encephalitis may have a negative viral load. More than half of the PCR tests on blood samples were performed on plasma or serum samples. Checking the EBV load in these matrices is recommended for the diagnosis and follow-up of nasopharyngeal carcinoma, while a viral load in whole blood may be more sensitive for other EBV-associated diseases [110,111].

EBV serology tests on blood samples seemed to be an interesting way to complement EBV PCR tests on CSF samples, especially in cases of primary EBV infection, where 25% of the CSF PCR tests are negative. In this situation, it may be interesting to repeat the lumbar puncture procedure after a few days, as is recommended in other types of viral encephalitis.

It should be noted that the analysis of virological data was impaired by the heterogeneity of the results: qualitative or quantitative PCR tests, in plasma or in whole blood samples, expressed in IU/mL or in copies/mL, with some serologies showing only one or two markers instead of the usual three. Finally, the methods used to determine the intrathecal antibody synthesis were not specified.

Regarding treatment, aciclovir was mainly used (69%), probably for the herpetic encephalitis. Ganciclovir was used in 30% of cases, corticoids were used in 53% of cases, and immunoglobulins were used in 15% of cases. Aciclovir was used for monotherapy in 23 cases, while ganciclovir was used in five cases, corticoids were used in nine cases, and immunoglobulins were used in two cases. Of the 39 patients treated with monotherapy (49%), two died. In some cases, drugs were used for multitherapy approaches (32 cases for aciclovir, 19 cases for ganciclovir, 33 cases for corticoids, and 10 cases for immunoglobulins), leading to 11 deaths. The overall mortality rate was 16% (15/97). Nevertheless, it is difficult to compare the efficacy of the different therapeutic options proposed here due to the lack of management standardization in the different cases. Furthermore, drug combinations were probably used for more severe or complicated cases.

To our knowledge, this is the first study to summarize data on EBV-related encephalitis cases. It highlights the limitations of EBV PCR testing on CSF samples, which is nevertheless the main laboratory test for diagnosis. It is therefore necessary to develop more efficient and/or complementary diagnostic tools. EBV serology tests on CSF samples were studied in seven cases and were positive except in one case of primary infection. This analysis is worth further exploration; determining an intrathecal secretion index (double ratio of serum/CSF EBV–serum/CSF albumin) that takes into account the permeability of the blood-brain barrier could improve interpretation. Moreover, the usual virological analysis should be standardized, with the systematic determination of the three markers IgG-VCA, IgG EBNA, and IgM VCA for serology testing, and for EBV PCR testing, the systematic quantification in IU/mL in the CSF and blood, and if possible, for both plasma and whole blood. In the best-case scenario, this might help to establish the diagnostic thresholds, or at the very least, to compare clinical situations (Appendix A).

This work also highlights the lack of treatment and standardization of practices, despite the attempts of a recent review to summarize therapeutic options [112]. The only drug approved for the treatment of EBV infectious diseases is rituximab, an anti-CD-20 antibody more often used in EBV-associated lymphomas. In this study, rituximab was only used in six cases, always in combination with other drugs, with two of the six patients dying. The other treatments are not approved for the EBV. Indeed, without untreated controls, the efficiency of such treatments is difficult to ascertain. It is now urgent to study this pathology and initiate clinical studies to make recommendations for the management of this disease, which, in this study, was fatal in 16% of cases. This rare disease will require multicenter studies, with possible contributions from several countries.

## Figures and Tables

**Figure 1 microorganisms-11-02825-f001:**
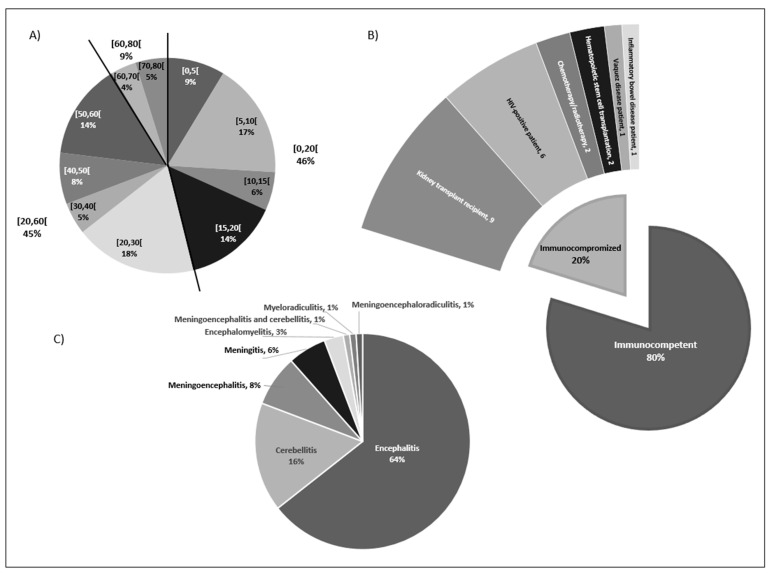
Distribution of patients by (**A**) age group ([0, 5[ = from 0 to 5 years excluded), (**B**) immunity characteristics (number = number of cases), and (**C**) depending on the final diagnosis.

**Figure 2 microorganisms-11-02825-f002:**
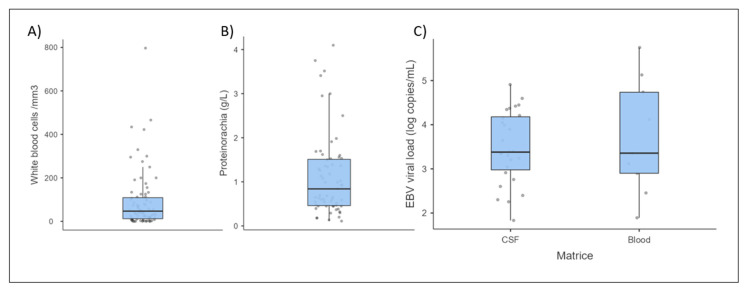
Distribution of (**A**) white blood cells in the cerebrospinal fluid (CSF) (N = 79), (**B**) protein concentrations in the CSF (N = 63), and (**C**) viral load in the CSF (N = 27) and blood (N = 9) for the reported cases of EBV encephalitis.Horizontal bars represent medians, boxes represent the first quartile (Q1 = 25%) and the third quartile (Q3 = 75%), while vertical bars represent the upper limit (Q3 + 1.5 × (Q3 − Q1)) and the lower limit (Q1 − 1.5 ∗ (Q3 − Q1)).

**Table 1 microorganisms-11-02825-t001:** Countries and references of the recorded cases.

Country	Number of Cases/Country	Continent	Number of Cases/Continent	Ref.
Italy	11	Europe	43	[16,17,60,63,64,66,69,71,77,89,91]
Spain	6			[10,34,45,50,68,92]
Germany	5			[18,46,57,72,95]
Belgium	3			[30,33,62]
France	3			[31,35,94]
Austria	3			[42,43,86]
United Kingdom	3			[36,44,85]
The Netherlands	2			[20,23]
Portugal	2			[28,88]
Norway	1			[25]
Croatia	1			[54]
Malta	1			[96]
Hungary	1			[79]
Japan	9	Asia	38	[26,37,41,67,70,81,83,93,101]
China	8			[13,29,47,52,58,73]
India	8			[21,39,51,80,87,107]
Turkey	4			[19,22,24,76]
South Korea	3			[53,65,90]
Taiwan	2			[75,100]
Iran	1			[14]
Thailand	1			[84]
United States	11	North America	11	[11,15,27,82,98,102,103,104,105,106]
Brazil	5	South America	7	[40,49,74,78,99]
Columbia	1			[56]
Ecuador	1			[55]
Australia	6	Australia	6	[12,32,38,48,59,61]
Niger	1	Africa	1	[97]

**Table 2 microorganisms-11-02825-t002:** Cerebrospinal fluid features of the reviewed cases of EBV encephalitis.

	Number of Documented Cases	Number of Positive Cases or > Reference Value (%)	Median Value	Highest Value
Proteins (g/L)	86	52	0.84	4.1
White blood cells (cells/mm^3^)	79	66 (83%)	47	797
Lymphocytosis	46	40 (87%)		
Neutrophil polynucleosis	46	6 (13%)		
EBV PCR	70	61 (87%)		
qPCR (copies/mL)	27		2400	81,200

**Table 3 microorganisms-11-02825-t003:** EBV PCR results in cerebrospinal fluid (CSF) and blood according to the EBV serological status.

EBV Serological Status	Number of Cases	Number of EBV PCR in CSF	Number of Positive EBV PCR in CSF	Number of EBV PCR in Blood	Number of Positive EBV PCR in Blood
**Primary infection**	36/75 (48%)	28/36	21/28 (75%)	11/36	9/11 (81%)
**Past infection**	21/75 (28%)	14/21	14/14 (100%)	6/21	2/6 (33%)
**Inconclusive**	18/75 (24%)				

**Table 4 microorganisms-11-02825-t004:** Imaging features of the reviewed cases of EBV encephalitis.

	N	%
**Abnormal images consistent with encephalitis**		
Abnormal MRI/performed MRI	61/88	69.3
Consistency between CT scan and MRI	12/27	44.4
Normal CT scan and abnormal MRI	14/27	51.9
**Localizations**		
Cerebellum	19	
Basal ganglia	12	
Frontal lobe	8	
Cerebral cortex	7	
Thalamus/hypothalamus	6	
Meninges	6	
Occipital lobe	5	
Temporal lobe	5	
Parietal lobe	4	
Diffuse	4	
Brainstem	4	
Corpus callosum	4	
Hippocampus	1	
**Electroencephalogram**		
Number performed	40	
Normal	7	17.7
Diffuse slowing	19	47.5
Discharges	11	27.5
Diffuse slowing + discharges	1	

N = number of cases, CT = computed tomograph; MRI = magnetic resonance imaging.

**Table 5 microorganisms-11-02825-t005:** Treatment reported for EBV encephalitis.

(Val) Acyclovir	(Val) Ganciclovir	IVIg	Corticoids	Total	Number of Deaths (%)
x				23	1 (4)
	x			5	0
		x		2	0
			x	9	1 (11)
x		x		2	1 (50)
x			x	19	3 (16)
x		x	x	1	0
	x		x	5	2 (40)
	x	x	x	4	0
x	x			5	1 (20)
x	x	x		1	1 (100)
x	x		x	2	2 (100)
x	x	x	x	2	1 (50)
**55**	**24**	**12**	**42**	**80**	**13**

x = administrated drug(s), IVIg = Intravenous immunoglobulins.

## Data Availability

The data presented in this study are openly available in PubMed: https://pubmed.ncbi.nlm.nih.gov/.

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
