# Peer review of "Epstein-Barr Virus Encephalitis: A Review of Case Reports from the Last 25 Years"

_microorganisms, 2023, doi:10.3390/microorganisms11122825_

Round 1

Reviewer 1 Report

Comments and Suggestions for Authors

Authors make a review on Epstein-Barr virus encephalitis: methods of detection, recoded cases and the possible treatments

The topic is interesting but i have major concern

Abstract

Rewrite the following to make it more digestible for reader. Negative EBV polymerase chain reaction (PCR) in CSF occurred in primary EBV infections but not in past infections.

This work highlights the limitations of EBV PCR in CSF despite being the main laboratory diagnostic test as well as the need for more accurate diagnostic tools, approved treatments, and standardized practices. Please need more explanation. I think PCR is an accurate approach.

Introduction

Epstein-Barr virus (EBV) is highly prevalent with more should be changed into Epstein-Barr virus (EBV) is a highly prevalent with more

Materials and methods

I think adding information for the previous used methods for detection of virus with their references. To avoid repetition authors could add a table illustrate method of diagnosis and references. Because the presentation section and data visualization of results is vague, I think adding a table in this way in the results section could be also made

Table 1 Countries and references of the reviewed cases should be changed into Countries and references of the recorded cases

Discussion

I usually ask about this point Despite being the three most common etiology of viral encephalitis, EBV encephalitis is a rare and poorly documented disease with very few publications available on this subject. How most common and rare please add an obvious explanation.

EBV PCR in CSF was the most frequent test performed after WBC and protein concentration in CSF and was positive in the vast majority of cases (87%). Thus, EBV PCR in CSF appears to be crucial for the etiological diagnosis of EBV encephalitis. This sentence interferes with the sentence in abstract (This work highlights the limitations of EBV PCR in CSF despite being the main laboratory diagnostic test as well as the need for more accurate diagnostic tools, approved treatments, and standardized practices. Please need more explanation. I think PCR is an accurate approach). The PCR is specific method because it uses specific primers.

EBV serology in blood seemed to be interesting to complement EBV PCR in CSF. I think serology is of not value due to cross reaction and it is not accurate like PCR

Comments on the Quality of English Language

The review needs moderate English editing

Author Response

Authors: We thank the reviewer, for their suggestions and the time spent to improve our manuscript. Please find attached the modified version and below our answer to the editor and the reviewer.

We hope we have made all the corrections and answered all the questions of the editor and the reviewers. We hope that our article is now suitable for publication. If this is not the case we remain at your disposal to make further changes.

Reviewer 1

Authors make a review on Epstein-Barr virus encephalitis: methods of detection, recoded cases and the possible treatments

The topic is interesting but i have major concern

Abstract

Rewrite the following to make it more digestible for reader. Negative EBV polymerase chain reaction (PCR) in CSF occurred in primary EBV infections but not in past infections.

Authors: we agree with the reviewer that this sentence is not very easy to understad. We have rewritten it.

« Negative EBV polymerase chain reaction (PCR) in CSF occurred in primary EBV infections but not in past infections. »

Changed for

« When encephalitis occurred in the context of past EBV infection, all EBV PCR in CSF were positive. On the contrary negative EBV PCR in CSF occured only in the context of primary infections. »

This work highlights the limitations of EBV PCR in CSF despite being the main laboratory diagnostic test as well as the need for more accurate diagnostic tools, approved treatments, and standardized practices. Please need more explanation. I think PCR is an accurate approach.

Authors: We also believe that PCR is a specific, accurate and suitable approach.

In the abstract, this is not what we meant. EBV PCR in the CSF is a useful tool to help diagnose EBV encephalitis. It is, however, imperfect, and it would be interesting to find complementary tools.

We changed the abstract and we hope it better explain our point of view :

“This work highlights the limitations of EBV PCR in CSF despite being the main laboratory diagnostic test as well as the need for more accurate diagnostic tools, approved treatments, and standardized practices.”

Changed for

“This work highlights that EBV PCR in the CSF is today the main laboratory diagnostic test to diagnose EBV encephalitis. It is useful however imperfect. New complementary diagnostic tools, approved treatments, and standardized practices could improve patient management.”

Introduction

Epstein-Barr virus (EBV) is highly prevalent with more should be changed into Epstein-Barr virus (EBV) is a highly prevalent with more

We modified the text accordingly

Materials and methods

I think adding information for the previous used methods for detection of virus with their references. To avoid repetition authors could add a table illustrate method of diagnosis and references. Because the presentation section and data visualization of results is vague, I think adding a table in this way in the results section could be also made

Authors: We are not sure we understand what Reviewer 1 means by this comment.

We have tried to draw up a table showing which analyses are reported in which publications, regardless the analysis result is positive or negative.

If you think the table will help the reader we can add it and call it as shown below:

3.3. Biological data

3.3.1. Cerebrospinal fluid and brain analysis

Among the 97 selected cases, polymerase chain reaction (PCR) for EBV DNA detection in cerebrospinal fluid (CSF) was performed in 70 patients (Supplementary Table).

For our point of view, the information provided in this table is too limited to be of any real interest.

Table 1 Countries and references of the reviewed cases should be changed into Countries and references of the recorded cases

The title was modified accordingly

Discussion

I usually ask about this point Despite being the three most common etiology of viral encephalitis, EBV encephalitis is a rare and poorly documented disease with very few publications available on this subject. How most common and rare please add an obvious explanation.

Authors: we agree with the reviewer comment. We have rewritten the sentence.

« EBV encephalitis is one of the 3 to 5 most frequent causes of viral encephalitis, depending on the study. However, our literature search identified 97 clinical cases describing EBV-related encephalitis. This work aimed to study and summarize these rarely published cases. »

EBV PCR in CSF was the most frequent test performed after WBC and protein concentration in CSF and was positive in the vast majority of cases (87%). Thus, EBV PCR in CSF appears to be crucial for the etiological diagnosis of EBV encephalitis.

This sentence interferes with the sentence in abstract (This work highlights the limitations of EBV PCR in CSF despite being the main laboratory diagnostic test as well as the need for more accurate diagnostic tools, approved treatments, and standardized practices.

Please need more explanation. (I think PCR is an accurate approach). The PCR is specific method because it uses specific primers.

Authors: We also believe that PCR is a specific, accurate and suitable approach.

In the abstract, this is not what we meant. EBV PCR in the CSF is a useful tool to help diagnose EBV encephalitis. It is, however, imperfect, and it would be interesting to find complementary tools.

We changed the abstract and we hope it better explain our point of view :

“This work highlights the limitations of EBV PCR in CSF despite being the main laboratory diagnostic test as well as the need for more accurate diagnostic tools, approved treatments, and standardized practices.”

Changed for

“This work highlights that EBV PCR in the CSF is today the main laboratory diagnostic test to diagnose EBV encephalitis. It is useful however imperfect. New complementary diagnostic tools, approved treatments, and standardized practices could improve patient management.”

EBV serology in blood seemed to be interesting to complement EBV PCR in CSF. I think serology is of not value due to cross reaction and it is not accurate like PCR

We agree with the reviewer.

serology is much less specific than PCR. We meant here that, when EBV PCR in the CSF is negative and the serology shows a EBV primary infection, it is important to take a new CSF sample after a few days for a new EBV PCR analysis.

We have rewritten the discussion : .

« EBV serology in blood seemed to be interesting to complement EBV PCR in CSF. Indeed, in the case of EBV encephalitis resulting from past EBV infection, EBV PCR in CSF was always positive, whereas in primary infection, it was negative in 25% of cases. »

Was changed for

« EBV serology in blood seemed to be interesting to complement EBV PCR in CSF especially in cases of EBV primary infection, where  25% of the CSF PCR are negative. »

Reviewer 2 Report

Comments and Suggestions for Authors

Interesting article. Written in nice English. Authors should review the literature list, some journal names are spelled incorrectly.

It is also worth considering rephrasing some sentences, as numerous repetitions make reading difficult.

In the reviewer's opinion, the entry about very few publications on neurological complications of EBv results from too narrow setting of the study boundaries. It is worth considering whether the literature should be expanded to include journals that discuss EBv encephalitis, but have slightly broader titles.

Author Response

Authors: We thank the reviewer, for their suggestions and the time spent to improve our manuscript. Please find attached the modified version and below our answer to the editor and the reviewer.

We hope we have made all the corrections and answered all the questions of the editor and the reviewers. We hope that our article is now suitable for publication. If this is not the case we remain at your disposal to make further changes.

Reviewer 2

Interesting article. Written in nice English. Authors should review the literature list, some journal names are spelled incorrectly.

It is also worth considering rephrasing some sentences, as numerous repetitions make reading difficult.

In the reviewer's opinion, the entry about very few publications on neurological complications of EBv results from too narrow setting of the study boundaries. It is worth considering whether the literature should be expanded to include journals that discuss EBv encephalitis, but have slightly broader titles.

Authors: thank you for accepting to review our manuscript and for the compliments.

We extensively reviewed the references to eliminate mispelled one.

We tried to rephrase some sentences.

Thank you for your latest comment. In a future job we'll try to broaden our search with more terms. However, you will understand that for this study, this would require to complete redesign the entire work.  This will greatly delay the publication of this work, which has never been done before.

Comments included in the pdf text :

Abstract : Which population? General? Adults?

Authors : We thank the reviewer for this comment we added the missing information

Although uncommon, Epstein-Barr virus-related neurological disorders represent the seventh cause of infectious encephalitis in adults.

Introduction:

The statement that this is a small description of the effects is most likely due to the narrow settings of the pub med search results. If the word encephalitis was not included in the publication, it was not included. It is worth paying attention to this point more broadly and only then narrowing down your search.

Mazur-Melewska K, Breńska I, Jończyk-Potoczna K, et al. Neurologic Complications Caused by Epstein-Barr Virus in Pediatric Patients. J Child Neurol. 2016;31(6):700-708. doi:10.1177/0883073815613563

Doja A, Bitnun A, Ford Jones EL, et al. Pediatric Epstein-Barr virus-associated encephalitis: 10-year review. J Child Neurol. 2006;21(5):384-391. doi:10.1177/08830738060210051101

Authors : We agree with this statement. As explained higher in a future job we'll try to broaden our search with more terms. However, you will understand that for this study, this would require to complete redesign the entire work. This will greatly delay the publication of this work, which has never been done before except, for pediatric cases, in the publication you cite covering 1996 to 2006..

Discussion :

This is a description of the results, not a discussion

Authors: we changed the sentence to eliminate the data that should be in the results section and leave only the comparison with the literature.

« The patients were essentially immunocompetent and young subjects (in agreement with the number of EBV primary-infection) which is comparable to all-cause encephalitis cohorts such as the cohort of Granerod et al.108 in England involving 203 patients. »

Please correct the text for repetitions. 4 x EBV in one sentence. This is very tiring for the reader

Authors : We agree with the reviewer comment. The sentence have now 5 x EBV indeed the previous one have 10 x EBV ; We hope it looks better.

EBV PCR in blood was rarely included in the published cases, probably because it is not specific to the disease. Indeed, people with a high viral load in blood do not necessarily have EBV encephalitis, while those with EBV encephalitis may have a negative viral load. More than half of PCR in blood were performed on plasma or serum. EBV load in these matrix is recommended for the diagnosis and follow-up of nasopharyngeal carcinoma, while virla load in whole blood may be more sensitive for other EBV-associated diseases.114

Why not with bold?

Authors : The new sentence is :

« In this situation, it may be interesting to repeat the lumbar puncture after a few days, as is recommended in other viral encephalitis. »

References :

Le Maréchal M, Mailles A, Seigneurin A, et al. A Prospective Cohort Study to Identify Clinical, Biological, and Imaging Features That Predict the Etiology of Acute Encephalitis. Clin Infect Dis. 2021;73(2):264-270. doi:10.1093/cid/ciaa598

The name of journal is not correct

It is not the correct citation. The name of the journal is uncorrect:

Dyachenko P, Smiianova O, Kurhanskaya V, Oleshko A, Dyachenko A. Epstein-barr virus-associated encephalitis in a case-series of more than 40 patients. Wiad Lek. 2018;71(6):1224-1230.  (shortage Wars  is connected with the city)

Authors : The 2 references have been corrected

Round 2

Reviewer 1 Report

Comments and Suggestions for Authors

Dear authors thank you for your response I could accept the review article in its current form